# Controlling Algorithm of Reconfigurable Battery for State of Charge Balancing Using Amortized Q-Learning

Dominic Karnehm [1,*], Wolfgang Bliemetsrieder [2], Sebastian Pohlmann [1] and Antje Neve [1]

1   Electrical Engineering and Technical Informatics Department, University of the Bundeswehr Munich, 85577 Neubiberg, Germany; sebastian.pohlmann@unibw.de (S.P.); antje.neve@unibw.de (A.N.)
2   Electrical Power Systems and Information Technology, University of the Bundeswehr Munich, 85577 Neubiberg, Germany; wolfgang.bliemetsrieder@unibw.de
*   Correspondence: dominic.karnehm@unibw.de

**Abstract:** In the context of the electrification of the mobility sector, smart algorithms have to be developed to control battery packs. Smart and reconfigurable batteries are a promising alternative to conventional battery packs and offer new possibilities for operation and condition monitoring. This work proposes a reinforcement learning (RL) algorithm to balance the State of Charge (SoC) of reconfigurable batteries based on the topologies half-bridge and battery modular multilevel management (BM3). As an RL algorithm, Amortized Q-learning (AQL) is implemented, which enables the control of enormous numbers of possible configurations of the reconfigurable battery as well as the combination of classical controlling approaches and machine learning methods. This enhances the safety mechanisms during control. As a neural network of the AQL, a Feedforward Neuronal Network (FNN) is implemented consisting of three hidden layers. The experimental evaluation using a 12-cell hybrid cascaded multilevel converter illustrates the applicability of the method to balance the SoC and maintain the balanced state during discharge. The evaluation shows a 20.3% slower balancing process compared to a conventional approach. Nevertheless, AQL shows great potential for multiobjective optimizations and can be applied as an RL algorithm for control in power electronics.

**Keywords:** reconfigurable battery; neuronal network; SoC balancing; reinforcemnt learning; amortized Q-learning

## 1. Introduction

The balancing of the State of Charge (SoC) in battery packs is one of the main challenges in the field of Electrical Vehicles (EVs). Therefore, many types of balancing methods have been proposed. It has to be distinguished between active and passive balancing methods. Passive methods are power-loose methods. For this purpose, battery cells with a higher SoC transfer energy through a shunt resistor to discharge the cell [1]. This method is cost-effective in implementation. However, it reduces the efficiency of the battery pack. Active equalization circuits can be used in battery packs to transfer energy from cells with higher charge to those with lower charge [2–4]. Van et al. [5] propose an SoC balancing algorithm for battery cells connected in series. Therefore, a modified bidirectional cuk converter is used to transfer energy from cells with a higher SoC to cells with a lower one. The usage of this algorithm has been proven during the discharge and relaxation of the battery pack. In the literature, different Reinforcement Learning (RL) algorithms have been discussed to optimize the system operation parameters. For instance, balancing the voltage of the capacitor and the thermal stress [6] of a modular multilevel converter (MMC) or the optimization of the efficiency of the DC-DC converter of the dual active bridge (DAB) [7,8].

Reconfigurable batteries allow for the active balancing of different battery parameters, such as voltage [9], State of Temperature (SoT) [10], and SoC [11–13]. To address the

problem of SoC balancing in reconfigurable batteries and the modular multilevel inverter (MMI), various algorithms have been suggested. Several centralized algorithms have been proposed to balance the SoC during battery relaxation [12,13] and during the charge and discharge of the reconfigurable battery [11]. To reduce the communication required between the main controller and the indivudual modules, decentralized algorithms have been introduced [14–17]. Furthermore, machine-learning-based algorithms have shown possible benefits in the field of controlling reconfigurable batteries. Jiang et al. [18] propose an RL Deep Q-Network (DQN) to control a reconfigurable battery using a three-switch topology Battery Modular Multilevel Management (BM3) [19]. Stevenson et al. [20] use a DQN to reduce the imbalance of the SoC and the current of a reconfigurable battery with four battery cells. The authors improved the potential to increase economic viability and enhance the battery operation, which is crucial in the context of sustainability of EVs.

Mashayekh [21] introduces a decentralized online RL algorithm to control an SoC balanced modular reconfigurable battery converter based on the topology of the half-bridge converter. The focus of this method is to reduce the communication between the controller and the modules as well as to reduce the intercommunication between the modules to a minimum. Therefore, an algorithm based on game theory is implemented, where each module tries to maximize the reward of itself. Caused by the design of the reward function of each module, the reward of the entire system increases if each individual module maximizes its reward. The algorithm shows high usability for a balanced system. In the case of an unbalanced initial state, the authors suggest the implementation of other algorithms to balance first, followed by the usage of the decentralized algorithm to reduce the communication requirements during the balanced state.

Yang et al. [22] propose an online-learning DQN algorithm to balance the SoC of a reconfigurable battery with 64 cells and a predefined voltage output. For this, the authors used a neural network with one hidden layer.

The multiparameter balancing of the SoC and the State of Health (SoH) with an offline DQN algorithm have been introduced in [23]. The authors have implemented a DQN algorithm on a direct current (DC) reconfigurable battery using a half-bridge topology with 10 modules.

To balance the SoC of an Alternating current (AC) reconfigurable battery, a Q-learning algorithm has been proposed in [24]. The algorithm is restricted by the number of controllable modules in the system. However, the Q-learning algorithm has shown potential for controlling reconfigurable batteries. In comparison to multiparameter optimization, the limitations of the Q-learning algorithm regarding the possible number of controlled modules have to be faced. This work proposes an algorithm based on Amortized Q-learning (AQL) that addresses this problem. Additionally, the proposed algorithm allows for the combination of classical balancing algorithms and machine learning approaches. As a result, the positive aspects of both approaches can be utilized. Among others, this includes the safety of a classical approach and the flexibility and adaptation possibilities of machine learning methods. Furthermore, the algorithm can be applied to AC and DC reconfigurable batteries. To the best of the authors' knowledge, this work introduces the first reinforcement learning algorithm that enables the combination with classical algorithms to control reconfigurable batteries with variable voltage levels.

## 2. Reconfigurable Battery

This paper discusses two topologies of MMC for reconfigurable batteries: the half-bridge [9] and the BM3 [19] converter. In Figure 1, the circuits for both converters are shown. For a reconfigurable battery, multiple modules are interconnected with each other. A half-bridge converter module includes a battery and two MOSFET switches, $S_1$ and $S_2$, and can be switched to serial and bypass modes. A BM3 module includes a battery and three MOSFETs, $S_1$, $S_2$, and $S_3$, and it can be switched into serial, parallel, or bypass mode.

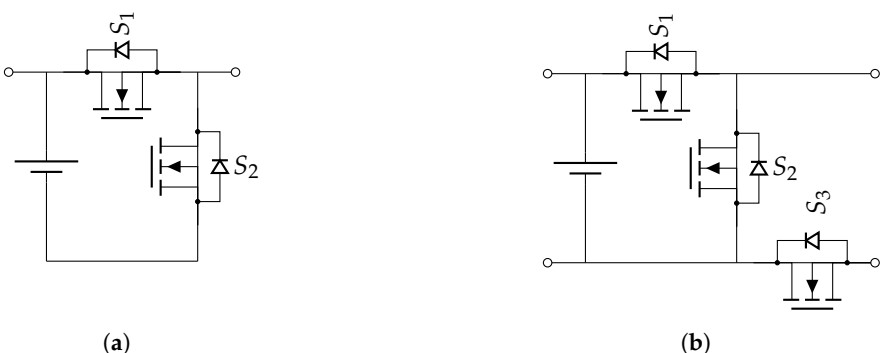

**Figure 1.** Electrical circuit of a half-bridge converter module (**a**) and a BM3 converter module (**b**).

Table 1 shows the three switchable modes. The two states of the MOSFET switches are defined as follows: on-state with a fixed resistance of $0.55\,\text{m}\Omega$ and off-state with an infinite resistance. The resistance value of the MOSFET is based on the resistance $R_{on}$ of the switches installed on the device under test (DUT) during the experimental evaluation.

**Table 1.** Switch states of the MOSFETs for BM3 and half-bridge, modeled as electrical resistors.

|  | $S_1$ | $S_2$ | $S_3$ |
|---|---|---|---|
| *Half-Bridge* |  |  |  |
| Bypass | on | off | - |
| Series | off | on | - |
| *BM3* |  |  |  |
| Bypass | on | off | off |
| Series | off | on | off |
| Parallel | on | off | on |

In this work, the SoC is calculated by Coulomb counting, where the *SOC* at the time step $t + \Delta t$ is defined as:

$$SOC(t + \Delta t) = SOC(t) - \int_0^{\Delta t} \frac{i_B}{Q_0} dt, \tag{1}$$

where $i_B$ is the current of the battery, $Q_0$ the capacity of the cell, and $SOC(0)$ is the initial SoC of the battery cell. Van et al. [5] explains that the balance of the SoC is achieved by equalizing the SoC of each cell so that the difference between the average SoC of each cell and the general average SoC is minimized. It is defined as follows:

$$\min \sum_{k=0}^{N} \left( SOC^k - M_{SOC} \right)^2$$

$$M_{SOC} := \frac{1}{N} \sum_{k=0}^{N} SOC^k \tag{2}$$

## 3. Reinforcement Learning Model

Reinforcement Learning (RL) [25] is one of the three main paradigms of machine learning. The others are supervised and unsupervised learning. In Reinforcement Learning (RL), an agent learns to maximize the cumulative reward by interacting with an environment. Due to this, training data is not necessary. On the contrary, an environment is essential during the process of training. An RL problem is defined as a Markov Decision Process (MDP) containing a set of environmental states $S = \{\mathbf{s}_1, \mathbf{s}_2, \dots, \mathbf{s}_n\}$, possible actions by the agent to interact with the environment $A = \{\mathbf{a}_1, \mathbf{a}_2, \dots, \mathbf{a}_m\}$, and the reward function $r(\mathbf{a}, \mathbf{s})$. Figure 2 shows the training process by modeling the interaction between an agent and the environment. The agent chooses an action $\mathbf{a}_t$ based on the current state $\mathbf{s}_t$. Depending on

the state and action, the environment returns a reward $r_t := r(\mathbf{a}_t, \mathbf{s}_t)$, determined by the reward function, and the next state $\mathbf{s}_{t+1}$ [26,27].

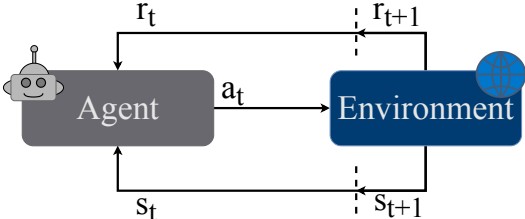

**Figure 2.** Interaction between agent and environment in reinforcement learning [25].

Generally, the actions an agent can perform are consistent across all states. A vector **a** represents an element of the action space *A* [26]. The objective of the proposed algorithm is to combine control methods and the possibilities provided by an RL algorithm. For the classical approach of a DQN, the neural network takes the state of the environment as input and outputs the action to take. This method limits the possibility of large discrete action spaces and restricts the action space. For the use case of operating a reconfigurable battery, the control of the output voltage is necessary. Additionally, the option of limiting the use of single modules is necessary for safety. Van de Wiele et al. [28] introduced Amortized Q-learning (AQL) an RL algorithm for enormous action spaces. This approach applies Q-learning to high-dimensional or continuous action spaces. The costly maximization of all actions **a** is replaced by the maximization of a smaller subset of possible actions sampled for a learned proposal distribution.

*3.1. State Space*

The state space *S* is defined by the normalized difference to the minimal SoC in the reconfigurable battery:

$$SOC_{diff}^k := SOC^k - \min(SOC) \tag{3}$$

$$SOC_{norm}^k := \frac{SOC_{diff}^k}{\max\left(SOC_{diff}\right)} \tag{4}$$

$$\mathbf{s} := \left[ SOC_{norm}^1, SOC_{norm}^2, \ldots, SOC_{norm}^N \right], \tag{5}$$

where $SOC^k$ is the SoC of the cell $k$, $SOC_{diff}^k$ is the difference from the minimum SoC $\min(SOC)$ of the system, and $SOC_{norm}^k$ is the difference normalized by the min-max scaler. The normalization enables the applicability in the case of large deviations between the SoC as well as in the case of battery cells with almost optimal balance.

*3.2. Action Space*

The action space *A* describes all possible actions **a**:

$$A := \left\{ \mathbf{a}^1, \mathbf{a}^2, \ldots, \mathbf{a}^W \right\}, \tag{6}$$

where *W* is the number of all possible actions. An action is defined as the switching states *m* of the modules of the reconfigurable battery pack.

$$\mathbf{a} := \left[ m^1, \quad m^2, \quad , m^N \right], \tag{7}$$

where $N$ is the number of modules in the system. The switching state $m^k$ of the module $k$ is described as follows:

$$m^{(k)} = \begin{cases} 0 & \text{if Bypass} \\ 1 & \text{if Serial} \\ 2 & \text{if Parallel} \end{cases} \tag{8}$$

Furthermore, the action space $A$ contains the subsets $A_V^{v(t)}$ to control the voltage levels. The number of battery cells switched to serial mode in order to generate the required voltage level at time $t$ is $v(t)$. Additionally, $A_C$ can restrict the action space $A$. The set $A_C$ contains excluded actions. It is possible to disable a collection of defined actions to allow for the combination of algorithmic and machine-learning-based control. An exemplary cause for such a restriction can be a broken MOSFET switch or a battery cell that is about to overheat. Consequently, the action space $A_t$ at time $t$ can be defined as:

$$A_t = A_V^{v(t)} - A_C. \tag{9}$$

### 3.3. Reward Function

The environment provides the reward, denoted $r_t$, to the agent. When the agent is in state **s** and takes action **a** at time $t$, the reward they receive is expressed as $r(\mathbf{s}_t, \mathbf{a}_t)$. The reward function used by the agent is defined as follows:

$$r^*(\mathbf{s}_t, \mathbf{a}_t) := [\max(SOC_t) - \min(SOC_t)] - $$
$$[\max(SOC_{t+1}) - \min(SOC_{t+1})] \tag{10}$$
$$r(\mathbf{s}_t, \mathbf{a}_t) := \gamma \cdot \max(0, r^*(\mathbf{s}_t, \mathbf{a}_t)), \tag{11}$$

where $\gamma$ is a bias value to increase the effect of the reward during training. It ensures that the reward is not too small and does not interfere with the optimization of the neural network.

### 3.4. Learning Algorithm

The Q-learning function used for training input to update the parameters $\theta^Q$ of the neural network is defined as follows [26]:

$$Q_{new}(\mathbf{s}_t, \mathbf{a}_t) := (1 - \alpha) \cdot Q(\mathbf{s}_t, \mathbf{a}_t) + \alpha \cdot r_t, \tag{12}$$

where $\alpha$ is the learning rate of the Q-learning approach $\alpha \in [0; 1]$ .

### 3.5. Neural Network and Training

Van de Wiele et al. [28] introduced AQL learning. All possible actions must be computed in methods such as Q-learning or DQN. AQL samples a defined subset of actions and maximizes the reward of this subset. The input of the network includes the state $\mathbf{s}_t$ and the action space $A_t$ of the reconfigurable battery as well as the actions $A_t$. The action $\mathbf{a}_t$ with the highest Q-value is identified by the function *argmax*. Each row of the input batch is defined as $[\mathbf{s}_t, \mathbf{a}_t^i]$, where $\mathbf{a}_t^i \in A_t$. The batch size is given by $B$, while $N$ describes the number of cells. Figure 3 shows the architecture of the proposed neural network.

The training algorithm that has been implemented is described in Algorithm 1, providing a detailed step-by-step procedure of the learning process.

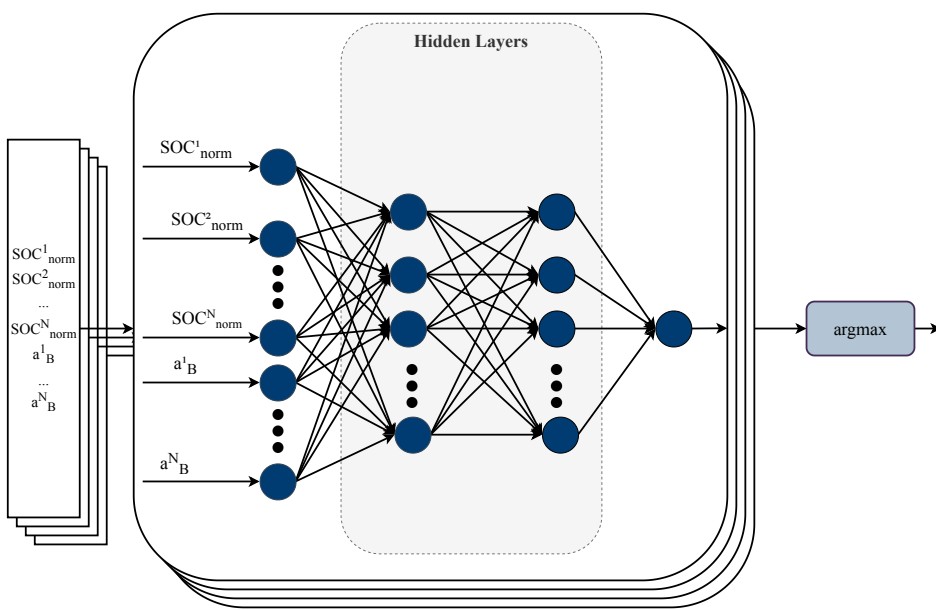

**Figure 3.** Architecture of neuronal network.

---

**Algorithm 1** Amortized Q-learning (AQL) training

---

1: **procedure** AQL TRAINING(Network parameters $\theta^Q$, exploration propability $\epsilon$, epochs $E$, epoch length $T$)
2:     Initialize local replay buffer $R$
3:     **for** $e \rightarrow 1 \cdots E$ **do**
4:         **for** $t \rightarrow 1 \cdots T$ **do**
5:             Observe state $\mathbf{s}_t$
6:             **if** random() $\leq \epsilon$ **then**
7:                 $a_t := a_t^R \in A_t$                                    ▷ Select $a_t$ at random
8:             **else**
9:                 $\mathbf{a}_t := \arg\max_{\mathbf{a} \in A_t} Q(\mathbf{s}_t, \mathbf{a}; \theta^Q)$
10:             **end if**
11:             $r_t, \mathbf{s}_{t+1} \leftarrow \text{environmentStep}(\mathbf{s}_t, \mathbf{a}_t)$
12:             $R_t \leftarrow (\mathbf{s}_t, \mathbf{a}_t, r_t, \mathbf{s}_{t+1})$
13:         **end for**
14:         Train $\theta^Q$ based on $R$ as defined (12)
15:         Reset *environment* to random state
16:     **end for**
17: **end procedure**

---

## 4. Description of Environment and Model

The neural network training and implementation are described below. A converter system with 12 modules is assumed for training and validation.

### 4.1. Training Environment

As an environment for training the models, a simulation of the BM3 converter introduced in [29], developed in Python using the numba framework as a just-in-time (JIT) compiler, is utilized. This framework allows for the simulation of a reconfigurable battery with the BM3 topology and the half-bridge topology, as seen in Figure 1.

### 4.2. Model Implementation and Training

This work implemented a Feedforward Neural Network (FNN) with three hidden layers and the rectified linear unit (ReLU) as an activation function. Table 2 shows the

architecture of the network. It is implemented in Python 3.8 utilizing the machine learning library PyTorch 2.0.1.

**Table 2.** Architecture of the model.

| Layers | Model |
|---|---|
| Input Layer | Dense (24) |
| Hidden Layer 1 | Dense (128) |
| | ReLU |
| | Dropout(0.1) |
| Hidden Layer 2 | Dense (64) |
| | ReLU |
| | Dropout (0.1) |
| Hidden Layer 3 | Dense (32) |
| | ReLU |
| | Dropout (0.1) |
| Output Layer | Dense (1) |

Figure 4 illustrates the reward for episodes during the training of both topologies. The hardware used to train the model was a PowerEdge R750xa (Dell, Round Rock, TX, USA) with an A40 GPU (Nvidia, Santa Clara, CA, USA) and an Xeon Gold 6338 CPU (Intel, Santa Clara, CA, USA). The total training times for the topologies half-bridge and BM3 come to 16.4 h and 47.0 h. Training requires a simulation due to the high number of training runs. A real-world setup could not archive the required number of runs in an acceptable training time. For evaluation during training, nine random initial SoCs are set for each battery cell. To ensure the reproducibility of the evaluation, a random seed is used. The evaluation reward is the sum of the reward, as defined in (11), of 0.1 s simulation time with a step size of $\Delta t = 10^{-5}$ s. Due to the complexity of action spaces, the training of both models does not have to take the same amount of epochs. Accordingly, 3000 and 10,000 epochs took place for the half-bridge and BM3 topology.

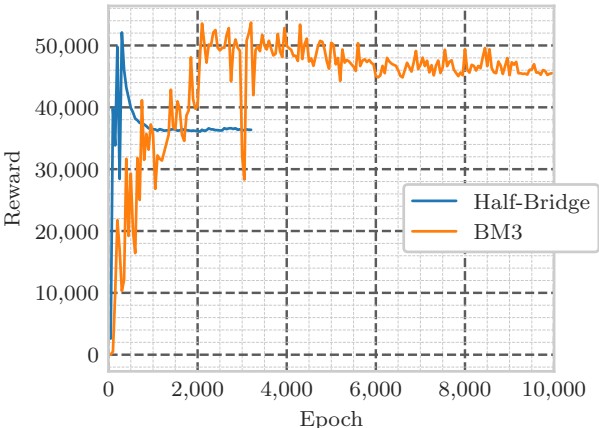

**Figure 4.** Reward over the training of the model for Half-Bridge (blue) and BM3 (orange) controlling.

## 5. Experimental Analysis

The usability of the proposed algorithm is analyzed and discussed based on the results of the experiments for the different scenarios:

- The simulative balancing of a 12-cell BM3 converter system.
- The experimental evaluation of results with a 12-cell half-bridge converter system and comparison with the balancing algorithm proposed by Zheng [30].

### 5.1. Simulative Evaluation

The simulation is utilized to analyze the practicability of the proposed algorithm in the field of BM3 converters. Figure 5 shows the SoC (a), voltage, and current (b), as well as the switching states (c) during a discharge interval over 10 s with a step size of $\Delta t = 10^{-4}$ s.

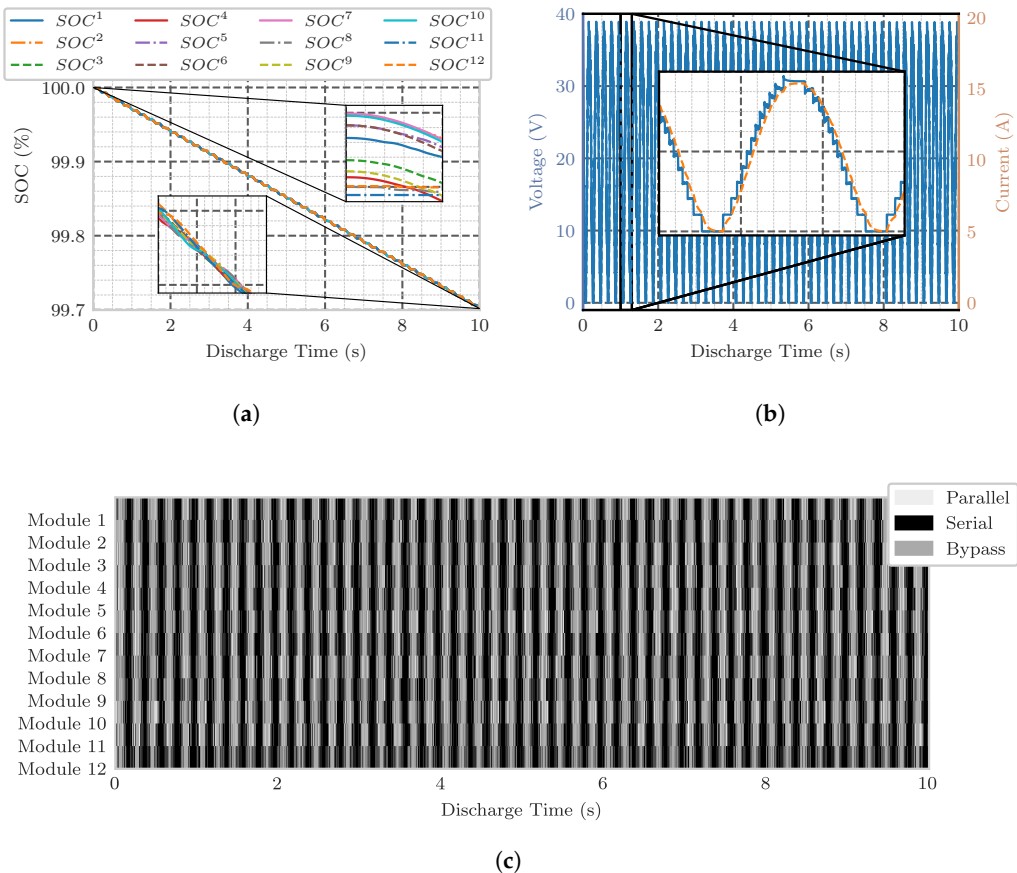

(a)                                                                      (b)

(c)

**Figure 5.** Simulated SoC (**a**), voltage (blue) and current (orange) (**b**), and states (**c**) of a BM3 converter over time using proposed AQL algorithm for 10 s and a step size for $\Delta t = 10^{-4}$ s.

Figure 5a illustrates the process of balancing the SoC. It shows a discharge of approximately 0.3% SoC of 12 BM3 modules. The zoom-in on the figure highlights the SoC balancing of the cells. It shows the unbalanced SoCs at the beginning and, afterwards, the shift to each other in order to reach the balanced state. Voltage and current during discharge can be seen in Figure 5b. The stepwise generation of the overall voltage caused by the multilevel inverter can be detected. The trained model uses all possible switch states with serial, bypass, and parallel. In 65% of all time steps, at least one cell is switched to parallel mode. It can be conducted that the proposed model can be utilized to control a BM3 reconfigurable battery and balance the SoC over time.

### 5.2. Experimental Evaluation

In addition to the evaluation based on the simulation of the proposed AQL algorithm to balance a BM3 converter, the algorithm is also evaluated in an experimental setting. The setup is shown in Figure 6 and described as follows:

- DUT: 12-cell hybrid cascaded multilevel converter [30], a topology of interconnected half-bridge modules and an h-bridge converter, as a reconfigurable battery module;
- Raspberry Pi 4 (Raspberry Pi Foundation, Cambridge, UK) as the control unit;
- ThinkPad-P15-Gen-1 (Lenovo, Hong Kong, China) as the computing unit;
- 2× Load resistor: MAL-200 MEG (MEGATRON Elektronik, Munich, Germany) 10 Ω in series;

- 12× Battery cell simulator: NGM202 (Rohde and Schwarz, Munich, Germany) Power Supply.

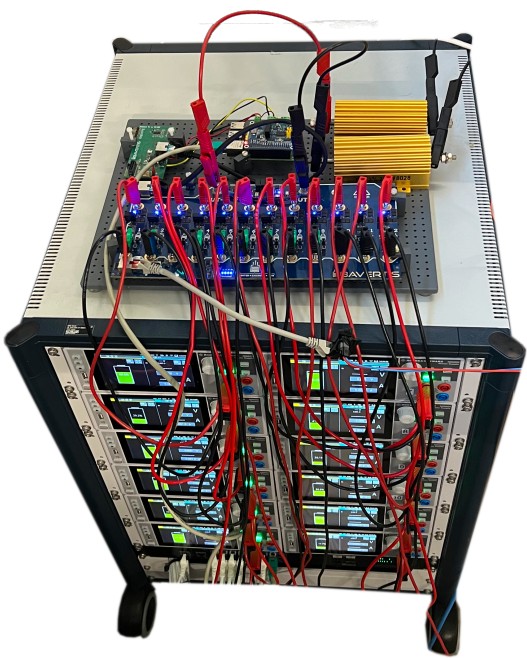

**Figure 6.** Experimental setup with a 12-cell hybrid cascaded multilevel converter as DUT.

A power supply unit is used for battery simulation to ensure a reproducible setup. Furthermore, this enables the establishment of a randomly chosen initial SoC between 100% and 70% of cells. The experimental ID is set as a seed for random value generation. During the experiment, cells are discharged from the initial SoC to 20%. The computing unit is dedicated to battery cell data processing and neural network execution. A Raspberry Pi 4 is available as the control unit, connected via the Controller Area Network (CAN) bus to the DUT. The control unit, the computing unit, and the battery cell simulators are connected over Ethernet for communication. The control unit sets the DUT switching states based on the determined actions by the computing unit.

The balancing process can be seen in Figure 7a. The discharge occurs for 100 s. The system reaches the balanced state after 71.9 s at a SoC of 54%. During validation, the balanced state is defined as a maximum SoC deviation of 1% within the cells with a capacity of 0.1 A h. Besides the discharge time itself, an approximately linear tapper of the discharge curves of the single cells can be detected.

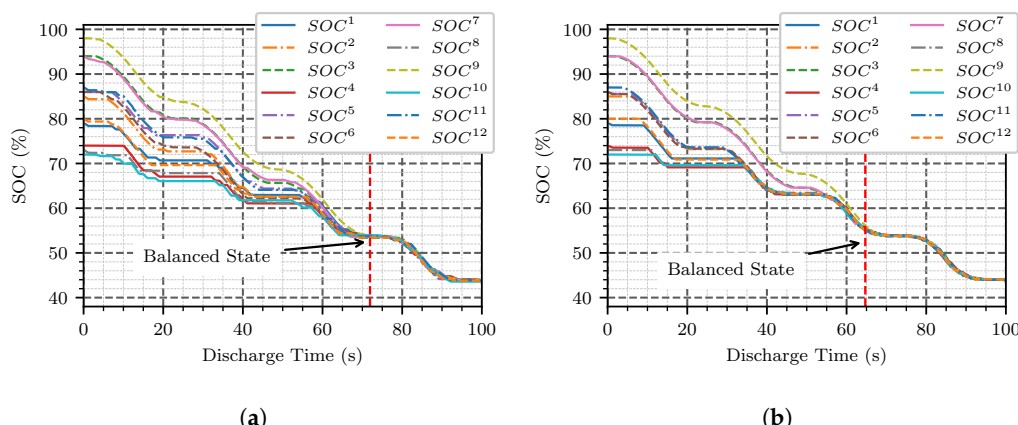

(**a**)                    (**b**)

**Figure 7.** SoC balancing using the proposed AQL algorithm (**a**) and switch-max algorithm (**b**); Balanced state after 71.9 s and 64.7 s (red) of discharge.

The balancing algorithm proposed by Zheng et al. [30] is also examined to evaluate the method proposed in this paper. The algorithm is defined as follows: a cell with a higher SoC can be discharged more. Those will be used preferably, and thus the difference in SoC is increased. Therefore, the *n* cells with the highest SoC are switched on, where *n* is the number of cells required to reach the requested voltage level. Due to this, the algorithm is following named switching-max. The results of the different balancing methods can be seen in Figure 7b. The initial SoC of each cell is set identically to ensure the same conditions for evaluating the proposed AQL algorithm. During the experiment, the switch-max algorithm reaches a balanced state and an SoC difference below 0.1 percent, after 64.7 s at an SoC of 55.4 percent.

For a valid evaluation of the proposed method and a discussion of the results compared to the algorithm proposed by Zheng et al. [30], the experiment was repeated 50 times with different randomly generated initial SoCs of each battery cell. The discharge time required to reach the balanced state of each experiment can be seen in Figure 8. In addition, it shows the mean (solid line) and standard deviation (shaded region) of both methods. It can be observed that the mean time required to balance the switching-max algorithm is 11.99 s or 20.3% faster compared to the proposed AQL algorithm. A mean balancing time of 70.9 s with a standard deviation of 11.9 s can be observed for the AQL method. Furthermore, the switching-max algorithm shows 58.9 s ± 10.0 s for balancing.

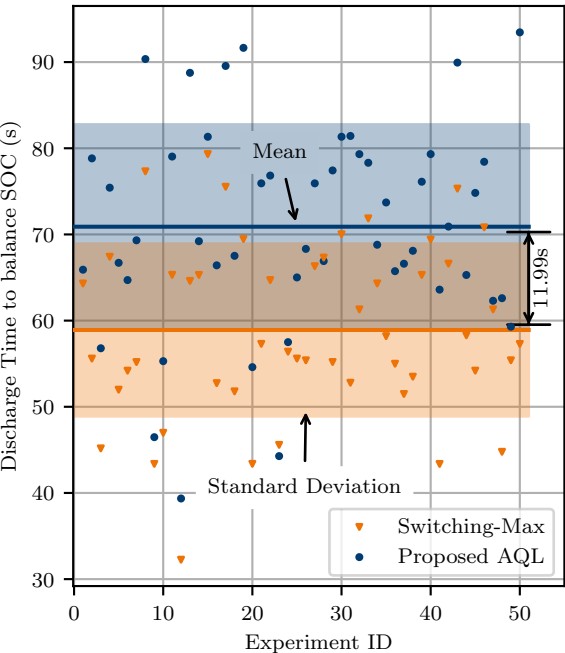

**Figure 8.** Comparasion of the required time to reach balanced state using the switching-max algorithm and the proposed AQL algorithm over 50 experiments; Area highlights the standard deviation of the mean discharge time (line).

## 6. Discussion

The general applicability of the proposed method is demonstrated by the evaluation based on the simulation of a BM3 converter and the experimental setup of the half-bridge converter. Both scenarios have shown the suitability for SoC balancing. Furthermore, after reaching the balanced state, the cells are steadily discharged. The algorithm can be utilized to balance an unbalanced state and to control the system during the balanced state. The computational cost is a significant component, mainly for the BM3 topology. With 12 modules, the voltage level of five cells connected in series has the highest number of valid configurations, resulting in a total of 19,448. Without any action constraints, the neural network must examine each configuration to determine the best option. The limitation of

the action space, proposed in (9), can help reduce computational cost and time. Furthermore, to decrease the number of switching operations per time step, a dynamic action space can be utilized. Therefore, the current switching configuration must be considered to limit the action space.

It is not possible to adequately compare the proposed method to other RL algorithms, such as [18,20,22], because they cannot be applied to different voltage levels. A separate neural network would be required for each voltage level. Comparison between the proposed AQL SoC balancing algorithm and the switching-max algorithm proposed by Zheng [30] shows a 12.0 s or 20.3% slower balance for the AQL algorithm. This conclusion was reached by comparing 50 individual experiments for each algorithm, as shown in Figure 8. However, the main goal of this work is to introduce an algorithm using a machine learning method that allows for the control of reconfigurable batteries with variable voltage output. The algorithm shows applicability for different types of topologies. Furthermore, the proposed algorithm enables the combination of classical algorithms with the proposed AQL algorithm by allowing dynamic action spaces as introduced in (9).

## 7. Conclusions and Outlook

This work introduced an Amortized Q-learning (AQL) model, a reinforcement learning algorithm, to balance the SoC of reconfigurable batteries. It is an improvement on the Q-learning algorithm introduced in [24]. Regarding the number of modules that can be controlled, the previous algorithm has limitations caused by memory requirements. Controlling more than seven modules was impossible due to a lack of memory during training. This paper addresses these limitations. The experimental validation has shown that training and controlling of 12 modules is assimilable. With the proposed algorithm, a combination of algorithmic and machine-learning-based control of the reconfigurable battery is possible. Among others, the control algorithm has to handle a high variety of use cases, be they individual cells that are idle to switch states due to broken MOSFET switches or for safety reasons, such as possible thermal runaway of cells. The effects of combining machine learning and the algorithmic approach are discussed in the studies.

The algorithm is evaluated in an experimental setup of a hybrid cascaded multilevel converter and a simulation of a BM3 converter. It suggests the applicability of the algorithm for the described scenarios. Figures 5a and 7a show the balancing process. A comparison with the algorithm proposed by Zheng [30] shows a 20.8% slower balancing. In addition, the proposed algorithm requires higher complexity and computational cost during execution. However, due to the lower switching time requirements, reconfigurable batteries as a DC source can be a field of application of the proposed algorithm. Therefore, in future works, it is necessary to evaluate whether AQL models can optimize multidimensional problems such as SoC balancing, loss reduction, and the thermal balancing of battery cells. A variant of the recurrent neural network (RNN) architecture is the choice for analysis. This type of neural network is developed to analyze time series data. This characteristic allows for the operation and controlling of time-related systems. The findings of the work by Yang et al. [23] will also be used in future work on the reconfigurable battery, which can be configured as a source of AC.

Furthermore, a comparison with the reinforcement learning algorithm that limits the communication required between different components of the reconfigurable battery proposed by Mashayekh [11] can indicate the various fields of application of the algorithms and the positive and negative aspects of both.

**Author Contributions:** Conceptualization, D.K. and A.N.; methodology, D.K.; software, D.K.; validation, D.K. and W.B.; formal analysis, D.K. and W.B.; data curation, D.K.; writing—original draft preparation, D.K.; writing—review and editing, W.B., S.P. and A.N.; visualization, D.K.; supervision, A.N.; and funding acquisition, A.N. All authors have read and agreed to the published version of the manuscript.

**Funding:** This research is funded by dtec.bw—Digitalization and Technology Research Center of the Bundeswehr, which we gratefully acknowledge. dtec.bw is funded by the European Union—NextGenerationEU. Further, we acknowledge financial support by the University of the Bundeswehr Munich.

**Institutional Review Board Statement:** Not applicable.

**Informed Consent Statement:** Not applicable.

**Data Availability Statement:** The data presented in this study are available on request from the corresponding author.

**Conflicts of Interest:** The authors declare no conflicts of interest.

## Abbreviations

The following abbreviations are used in this manuscript:

| | |
|---|---|
| AQL | Amortized Q-learning |
| AC | Alternating current |
| BM3 | Battery modular multilevel management |
| BMS | Battery Management System |
| DC | Direct Current |
| DUT | Device Under Test |
| DQN | Deep Q-Network |
| EVs | Electrical Vehicles |
| FNN | Feedforward Neural Network |
| MDP | Markov Decision Process |
| MOSFET | Metal-Oxide-Semiconductor Field-Effect Transistor |
| MMI | Modular Multilevel Inverter |
| MMC | Modular Multilevel Converter |
| RL | Reinforcement learning |
| SoC | State of Charge |
| SoH | State of Health |
| SoT | State of Temperature |

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
