# Peer review of "Controlling Algorithm of Reconfigurable Battery for State of Charge Balancing Using Amortized Q-Learning"

_batteries, doi:10.3390/batteries10040131_

Round 1

Reviewer 1 Report

Comments and Suggestions for Authors

In this work, Karnehm et al. propose a reinforcement learning-based method to balance the state of charge of reconfigurable battery systems. The proposed method is experimentally validated. In general, this work is novel and interesting. I suggest that it can be accepted after the following issues are properly addressed.

1. It is interesting to see the applications of machine learning in addressing battery management problems. This work features a reinforcement learning method. In practice, its training can be very costly and is usually conducted in a simulated environment (such as AlphaGo). In this regard, how do you train the reinforcement learning model and what is the cost?

2. Another important drawback of machine learning is its reliability. Although it gives good results in this work, how can we ensure safe battery control when it comes to actual battery management where numerous scenarios may be different from the lab tests? Alternatively, can engineers be informed when the machine learning model makes incorrect decisions?

3. It is interesting to see the reconfigurable battery systems. It seems we can make a battery like a “Movable type”. In addition to SOC, can we control the batteries to prolong the overall life of the system? If possible, please discuss if we can achieve this by extending your method.

Author Response

Comment 1

Review comment:
It is interesting to see the applications of machine learning in addressing battery management problems. This work features a reinforcement learning method. In practice, its training can be very costly and is usually conducted in a simulated environment (such as AlphaGo). In this regard, how do you train the reinforcement learning model and what is the cost?

Our answer:
The reviewer does have a good point to highlight the cost of the training. In section 4.2 (line 186 following) we added this information including the used simulation environment and the used hardware.

Comment 2

Review comment:
Another important drawback of machine learning is its reliability. Although it gives good results in this work, how can we ensure safe battery control when it comes to actual battery management where numerous scenarios may be different from the lab tests? Alternatively, can engineers be informed when the machine learning model makes incorrect decisions?

Our answer:
We agree with the author of this comment that reliability is one of the most important aspects when it comes to things like BMS. This is why we have introduced the possibility to limit the action space (as seen in eq. (9)). Standard DQN algorithms do not take this into account. With this feature we hope to increase the reliability of the algorithm and reduce the possibility of wrong decisions by the ML algorithm, because wrong decisions are already eliminated by the limited action space.

Comment 3

Review comment:
It is interesting to see the reconfigurable battery systems. It seems we can make a battery like a “Movable type”. In addition to SOC, can we control the batteries to prolong the overall life of the system? If possible, please discuss if we can achieve this by extending your method.

Our answer:
Since the submission of this paper, a new study has been published where the authors have shown a method to balance SOC and SOH using a DQN. The authors have used a half-bridge topology with ten modules. Furthermore, the reconfigurable battery is used as a DC source. This study does not include the possibility of limiting the action space, as we think it has an impact on reliability,  and also does not include the possibility of an AC source. However, it shows balancing of more than one parameter. In the paper, this new study has been added in the introduction and also in the outlook as an example that shows the possibilities of the general approach.

Reviewer 2 Report

Comments and Suggestions for Authors

Unfortunately, the paper is very difficult to understand because of the poor English used. The work is written in a very incorrect manner, and consequently, it is not clear at all. Moreover, there are many English mistakes. Therefore, the contribution of the work remains unclear. Moreover, it seems that the proposed method works worse than the one presented by Zheng [28].

Comments on the Quality of English Language

English very difficult to understand/incomprehensible

Author Response

We are deeply grateful for the reviewer’s insightful comments and have taken them into account in our revised manuscript. We have made substantial improvements to the language and clarity of the paper to enhance its readability.

The reviewer correctly pointed out that the proposed algorithm is slower than the one proposed by Zheng [29]. This is indeed acknowledged in the conclusion (line 303). Despite this, we believe our algorithm holds significant potential for future work, given its capacity to balance multiple parameters, as indicated in line 307. This potential was demonstrated for a deep Q-network in a paper published in December 2023 (Yang [23]).

We chose to compare our algorithm with Zheng’s [29] because it represents a state-of-the-art method for balancing SoCs for reconfigurable half-bridge batteries. While our results indicate that our method is currently not as efficient as the state of the art, we have chosen to publish our findings. We believe that the AQL method has potential as a machine learning control algorithm in the field of reconfigurable batteries, and we look forward to further exploring this potential in future work. We appreciate the opportunity to share our research and welcome any further feedback.

Reviewer 3 Report

Comments and Suggestions for Authors

1. Fix the text of the subtitle 4 - Description of Environment and Modle

2. In the conclusion, it is reported that the proposed algorithm requires higher complexity and computational cost during execution and is 12.0 s slower than the algorithm proposed by Zheng. In order not to derail the development of the AQL SoC balancing algorithm, it is good in this paper, rather than in future developments, to evaluate applications of this model to the optimization of other multidimensional problems.

Comments on the Quality of English Language

Minor revisions to the English language should be made. For example on the line 75 the should be a comma between "approaches and Providing".

Author Response

Comment 1

Review comment:
Fix the text of the subtitle 4 - Description of Environment and Modle

Our answer:
We thank the author for this comment. The subtitle have been changed to "Description of Environment and Model"

Comment 2

Review comment:
In the conclusion, it is reported that the proposed algorithm requires higher complexity and computational cost during execution and is 12.0 s slower than the algorithm proposed by Zheng. In order not to derail the development of the AQL SoC balancing algorithm, it is good in this paper, rather than in future developments, to evaluate applications of this model to the optimization of other multidimensional problems.

Our answer:
We do see also high potentials for future developments with this method. It is gorgeous to read, not only we see the potentials of this method. In future work, we will develop additional methods based on this work.

Round 2

Reviewer 2 Report

Comments and Suggestions for Authors

I understand the authors' intention to share their study, even if their proposed method is less performant than that of Zheng, with possible future improvements.

In any case, in my opinion, the issue lies in the English language. Indeed, I don't see any improvement compared to the previous version. For instance:

  • In line 99: "The on-state is considered with a conduction resistance of 0.55mΩ is considered."

  • In line 105: "is the currency of the battery."

  • In line 145: "where v(t) is the number of battery cells switched to serial mode."

  • In line 215: "Fig. 5c shows this. To generate the required voltage as a sinus wave, the cells are switched in 50% to serial, and in conclusion in 40% the cells are switched to bypass.

Comments on the Quality of English Language

The English requires thorough revision.

Author Response

We agree with the reviewer's points and have tried to improve the English language throughout the whole manuscript. We have also addressed the specific issues pointed out. 
We hope this improved the understanding of the work and the methods used.